# Advances in High-Energy-Resolution CdZnTe Linear Array Pixel Detectors with Fast and Low Noise Readout Electronics

**DOI:** 10.3390/s23042167

**Published:** 2023-02-15

**Authors:** Filippo Mele, Jacopo Quercia, Leonardo Abbene, Giacomo Benassi, Manuele Bettelli, Antonino Buttacavoli, Fabio Principato, Andrea Zappettini, Giuseppe Bertuccio

**Affiliations:** 1Department of Electronics, Information and Bioengineering (DEIB), Politecnico di Milano, Via Anzani 42, 22100 Como, Italy; 2National Institute of Nuclear Physics (INFN), Sezione di Milano, Via Celoria 16, 22133 Milan, Italy; 3Department of Physics and Chemistry (DiFC)—Emilio Segrè, University of Palermo, Viale delle Scienze, Edificio 18, 90128 Palermo, Italy; 4Due2lab s.r.l., Via Paolo Borsellino 2, 42019 Scandiano, Italy; 5IMEM-CNR, Parco Area delle Scienze 37/A, 43100 Parma, Italy

**Keywords:** CZT, CdZnTe, X-ray spectroscopy, Gamma-ray spectroscopy, semiconductor radiation detectors, nuclear microelectronics

## Abstract

Radiation detectors based on Cadmium Zinc Telluride (CZT) compounds are becoming popular solutions thanks to their high detection efficiency, room temperature operation, and to their reliability in compact detection systems for medical, astrophysical, or industrial applications. However, despite a huge effort to improve the technological process, CZT detectors’ full potential has not been completely exploited when both high spatial and energy resolution are required by the application, especially at low energies (<10 keV), limiting their application in energy-resolved photon counting (ERPC) systems. This gap can also be attributed to the lack of dedicated front-end electronics which can bring out the best in terms of detector spectroscopic performances. In this work, we present the latest results achieved in terms of energy resolution using SIRIO, a fast low-noise charge sensitive amplifier, and a linear-array pixel detector, based on boron oxide encapsulated vertical Bridgman-grown B-VB CZT crystals. The detector features a 0.25-mm pitch, a 1-mm thickness and is operated at a −700-V bias voltage. An equivalent noise charge of 39.2 el. r.m.s. (corresponding to 412 eV FWHM) was measured on the test pulser at 32 ns peaking time, leading to a raw resolution of 1.3% (782 eV FWHM) on the 59 keV line at room temperature (+20 °C) using an uncollimated ^241^Am, largely improving the current state of the art for CZT-based detection systems at such short peaking times, and achieving an optimum resolution of 0.97% (576 eV FWHM) at 1 µs peaking time. The measured energy resolution at the 122 keV line and with 1 µs peaking time of a ^57^Co raw uncollimated spectrum is 0.96% (1.17 keV). These activities are in the framework of an Italian collaboration on the development of energy-resolved X-ray scanners for material recycling, medical applications, and non-destructive testing in the food industry.

## 1. Introduction

Energy-resolved photon counting (ERPC) systems are object of increasing research interest due to their large impact in scientific, medical, and industrial applications, from computed tomography (CT) to inline inspection in food and pharma production chains, or material recycling processes [1,2,3,4]. Among compound semiconductors, detectors based on Cadmium Zinc Telluride (CdZnTe or CZT) are promising candidates for the development of next-generation ERPC systems, mainly competing with Cadmium Telluride (CdTe) detectors for room temperature operation, both being characterized by a wide band-gap, high stopping power, and the possibility to be realized in thick pixelated structures. CdTe detectors with Schottky contacts are currently able to guarantee high performances in terms of energy resolution. When combined with dedicated front-end electronics, CdTe detectors were demonstrated to have excellent energy-resolving capability, as required in high-rate and wide energy range applications (1–200 keV), such as in the detection of lightweight foreign bodies (plastics, insects, wood etc.) into products with thick or heavy packages (glass jar, tin can, etc.) [5,6,7,8] or in the identification and classification of plastics and electronic waste in material recycling processes [9,10].

On the other hand, due to the presence of a well-known performance instability in CdTe detectors, caused by bias-induced polarization phenomena [11,12], CZT detectors with quasi-ohmic contacts are considered to be more reliable solutions. CZT detectors allow the system to operate on a longer time scale without the need of resetting the high-voltage detector bias, with clear benefit from the reduction of the acquisition dead time associated with the high-voltage bias reset transient. Within this work, we will show that the energy resolution of CZT linear array detectors with quasi-ohmic contacts, when properly coupled with custom low-noise front-end electronics, can be dramatically improved, reaching comparable performance to what has been previously demonstrated with Schottky CdTe detector [7], in the application of room temperature ERPC systems, even with short pulse processing times (<100 ns). Using a 0.25 mm pitch CZT pixel, with 25 µm inter-pixel gap, and SIRIO charge sensitive amplifier [13,14] we measured an optimum full-width at half maximum (FWHM) of 576 eV (0.97%) on the ${}^{241}$Am 59.5 keV line at room temperature (+20 °C), with 222 eV FWHM (20.5 el. r.m.s.) on the test pulser. The detection system was tested using a ${}^{57}$Co source, showing a relative FWHM of 1.17% at +26 °C, which goes down to 0.66% if the low-energy tail is excluded from the peak fit.

It has been demonstrated that by employing CdTe and CZT and ultra-low noise electronics, it is possible to realize spectroscopic-grade X- ray imaging systems with electronic noise levels lower than 0.3 keV FWHM equivalent (Pulser linewidth) even at room temperature. This opens new applications for such high-Z semiconductor detectors demonstrating their capability to operate in the widest energy range, from soft X-ray (just above 1 keV) to hard X and rays (up to hundreds of keV) with high-energy revolving capabilities and at room temperature, unique conditions in the radiation detectors scenario. Such low noise not only makes spectroscopy at soft X-ray possible for such high-Z detectors but also allows to achieve an almost Fano-limited resolution at high energy, essential to resolve close-spaced spectral lines or to improve the details of continuous spectra in applications employing X-ray tubes.

## 2. Materials and Methods

The detector is based on a CZT crystal (2.95 × 10.40 × 1.10 mm^3^) grown by the boron oxide encapsulated vertical Bridgman (B-VB) technique [15]. The detector layout is characterized, on the back side (anode), by a linear array of pixels with 250 µm pitch and inter-pixel gap of 25 µm (Figure 1); an internal (500 µm width) and external guard-rings, surrounding the pixels, are used to minimize charge sharing events, and the surface and periphery leakage currents, respectively; the detector’s front-side (cathode) is a uniform planar electrode covering the whole surface, acting as entrance window of the incoming radiation for the following characterization, allowing a faster collection of holes generated by low-energy photons. The detector uses quasi-ohmic gold contacts ensuring low leakage current at room temperature (Figure 2) [16,17,18]. The pixel (anode) capacitance was evaluated to be ∼100 fF using Comsol Multiphysics simulation software [19], taking into account the total contributions due to the front-side contact (cathode), neighboring pixels, and guard rings.

The detector array was glued on one side of a test printed circuit board (PCB) and arranged to face the collecting pixel toward a narrow slit in the PCB. The other side of the PCB was used to accommodate the SIRIO high-speed CMOS charge sensitive amplifier (Figure 3) [13], working in pulsed reset configuration [20], optimized for minimum noise performance at deep sub-microsecond shaping times for low-capacitance detectors (<100 fF) [21,22,23]. The SIRIO preamplifier has an area occupation of 600 × 650 µm^2^ and has a conversion gain of 40 mV/fC, with a power consumption 15.3 mW/channel, including the integrated line-driver.

The connection between SIRIO and the CZT pixel was done using a wire wedge-bonding on the CSA input, and conductive silver epoxy on the CZT output. Adjacent pixels to the one under test have been grounded acting as an additional guard for the pixel under test. Thanks to an integrated test capacitance on the SIRIO CSA input, an accurate measurement of the electronic noise can be performed using a precision external pulser (Tektronix AFG3022C), which allows to evaluate different noise components and eventually the presence of unexpected noise contributions [24]. The SIRIO output was then buffered using an external low-noise line driver, to feed the input of two different commercial multi-channel analyzers: the XGLab Dante DPP [25] was used for the automated peaking-time sweep in fast low-statistics measurements (300 s acquisition time, with ≃300 counts on the 59.5 keV peak), while the Amptek PX5 [26] was used to characterize the system performance at the optimum filtering condition with high-statistics acquisitions. Both DPPs were configured to perform a symmetric, unipolar trapezoidal shaping [27,28] with 50 ns flat-top, which allows a reasonable compromise between shaping filter occupation time width and ballistic deficit caused by charge induction times [29].

In operating conditions, the front-side electrode is biased at −700 V using the CAEN N1471 high-voltage generator, combined with external passive filtering; the pixels are DC coupled to the preamplifiers’ inputs and the guard rings are grounded.

In estimating the energy resolution of the spectral lines, the Crystal Ball peak fit function, a heuristic piece-wise composed function based on a Gaussian function (on the right-hand side) and a power-law function for the low-energy tail (on the left-hand side) [30], has been combined with a step-like background fit function [31]. The width of the line is then evaluated on the fitted data as the full width of the peak at a height that is one-half of its maximum height above the underlying low-energy background [32].

## 3. Results

### 3.1. Spectroscopic Response

The spectroscopic response of the system was investigated by using ${}^{55}$Fe, ${}^{241}$Am, and ${}^{57}$Co uncollimated radioactive sources to test the performance of the detection system for different energy ranges without applying any type of charge sharing corrections. The results obtained with the different calibration sources are presented and discussed in the following sections.

#### 3.1.1. ${}^{55}$Fe

The resolving capability in the low-energy range was tested using a ${}^{55}$Fe radioactive source. Due to the presence of the close energy peaks at 5.9 keV and 6.5 keV, ${}^{55}$Fe spectra using CZT detectors have been acquired in the past cooling the detector at −30 °C or below [33,34]. In Figure 4 we report the spectrum acquired at room temperature (+20 °C) with our detection system at a peaking time of 1.4 µs, where the two peaks can be clearly resolved with a FWHM of 258 eV on the Mn K line. The FWHM of the pulser line is 189 eV, giving a lower limit energy resolution on the Mn K line of 224 eV FWHM (assuming a Fano factor of F = 0.1 from [33]). The noise threshold can be evaluated slightly below 1 keV. It can be observed a quite high background in the 1 keV to 5 keV region, which brings to a very low peak to background ratio of 6 ÷ 1. This can be explained by charge sharing between pixels and guard, and possibly to losses of signal charge when the photons interact close to the detector surface.

#### 3.1.2. ${}^{241}$Am

Using the Amptek PX5, the SIRIO-CZT system has been characterized at the optimum peaking time (1 µs). In Figure 5 the resulting ${}^{241}$Am spectrum of a 1-h acquisition is shown. The linewidth at room temperature of 576 eV FWHM (0.97%) on the 59.5 keV line confirms a state-of-the-art spectroscopic resolution performance, which lowers to 523 eV FWHM (0.88%) if the low-energy tailing is excluded, with a FWHM of 222 eV (20.5 el. rms) on the pulser line. Even with a non-optimal peak-to-background ratio of ≃20 ÷ 1, the low-noise performance of the acquisition systems allows to clearly resolve the Np L*α*, L*β* and L*γ* lines in the 10–22 keV energy range of the ${}^{241}$Am spectrum, visible in the inset of Figure 5. In the lower energy range (<6 keV), the presence of some spurious lines, resembling a scaled version of the ${}^{241}$Am spectrum, is observed; this effect, currently under investigation, could be due to a partial induction on the pixel under test from the signal charge collected in the neighbor electrodes (guard and pixels) that, even with a net zero area, could still cause a wrong detection in the MCA logic.

The Dante DPP has been used to acquire the FWHM on ${}^{241}$Am spectra at different shaping times, obtaining an optimum FWHM of 592 eV at 0.5 µs peaking time on the 59.5 keV line, corresponding to 0.99% of energy resolution. The presented results compare well with state-of-the-art CZT-based detection systems, where optimum FWHM of 540 eV has been recorded [35]. The minimum equivalent noise charge (ENC) of the SIRIO-CZT system, as measured on the electronic pulser using the Dante DPP at 0.8 µs peaking time is 20.7 electrons rms, corresponding to 217 eV in CZT. The optimum FWHM at 13.9 keV of 365 eV at 0.8 µs represents an excellent result at room temperature even if compared to performances obtained with high-resolution Si-based detectors [36].

#### 3.1.3. ${}^{57}$Co

The detection system was eventually tested using a ${}^{57}$Co source, to characterize the higher energy range. Acquisition with the SIRIO-CZT detection system using the Amptek PX5 DPP showed a relative FWHM of 0.96% (1.17 keV) at +26.3 °C and at 1 µs peaking time, which lowers to 0.66% (0.81 keV) if low-energy tail due to incomplete charge collection is excluded from the peak fit. The measured peak to background on the 122 keV line is 5 ÷ 1. The FWHM on the electronic pulser is 282 eV. As shown in the insets of Figure 6, the left-side tail is largely reduced for the low-energy peak at 14 keV, where a FWHM of 364 eV is measured.

### 3.2. Energy Resolution with Fast Signal Shaping

Using the Amptek PX5 DPP, 1-h acquisitions were performed with the ${}^{241}$Am source at the shortest available peaking times (50 and 100 ns), to further back the results of stability over time performance of the detection system even in presence of ballistic deficit pulse processing [37]. Measurements showed 774 eV FWHM on the 59.5 keV line (377 eV FWHM on the pulser) for the 50 ns peaking time and 744 eV FWHM (309 eV FWHM on the pulser) for the 100 ns peaking time acquisition. In Figure 7, it is shown that even at 32 ns peaking time with the Dante DPP, the energy resolution on the 59.5 keV line is 1.3% (782 eV FWHM), with 412 eV FWHM on the pulser line, a performance which is still among the best results present in recent scientific literature, obtained using longer shaping times in the µs range [38,39,40]. Using the ${}^{57}$Co source at 50 ns peaking time, the measured FWHM at 122 keV is 2.13 keV (1.75%) which is mostly limited by the significant low-energy tail that, at this fast shaping time, lowers the peak-to-background ratio down to 4 ÷ 1.

### 3.3. Time Stability

In order to evaluate the presence of the bias-induced polarization phenomena in the detector, a long measurement of 7 h was performed using a low input photon count-rate of ≃1 kcps, without depolarizing the detector and with intermediate acquisitions every hour. The acquired spectra are reported in Figure 8. The stability of the spectra in terms of charge collection efficiency and energy resolution is evident across the complete acquisition. In Figure 9 the 59.5 keV peak shift and FWHM degradation are reported with respect to a reference 5-minute acquisition performed at the start of the 7 h. It can be noted that the peak shift is contained with 0.45%, which determines a FWHM deterioration of less than 35%.

## 4. Discussion

A high-energy-resolution detection system based on a CZT detector and SIRIO CSA has been described and characterized using uncollimated radioactive sources in different energy ranges. The spectroscopic performance of the system allows reaching 258 eV FWHM (4.34%) on the 5.9 keV line, 576 eV FWHM (0.97%) on the 59.5 keV line, and 1.17 keV FWHM (0.96%) on the 122 keV line. As a result of this characterization at different energies, it is observed that a gap between the measured FWHM on the spectral lines and the expected performance obtained summing the pulser FWHM with the Fano noise contribution is present. This gap can be seen in detail at different shaping times for the 59.5 keV and the 13.9 keV lines in Figure 10, and is summarized in Table 1, where the excess contribution is quoted for the main experimental measurements.

This effect can be attributed mostly to the charge sharing on the uncollimated pixel, trapping noise, and ballistic deficit related to signal components associated to the holes, which are not accounted for in the pulser line width. These causes produce a characteristic left-hand-side tailing on the acquired spectral lines and contribute to a relatively poor peak-to-background ratio observed across all spectra. Such tailing effects are more prominent in the lines associated to high energy photons, which are stopped deeper inside the detector volume [42]. This allows inferring that the presented spectroscopic performance could be further improved by applying appropriate corrections in the spectral reconstruction using time coincidence analysis, or depth of interaction correction techniques as widely discussed in the scientific literature of detection systems with imaging capability [43,44]. Figure 10 also shows the decomposition of the experimental data in terms of series and parallel noise components, represented using black dashed lines. In Figure 11 the measured FWHM at different photon energies is divided into the percentual contributions arising from the Fano noise (pink), the electronic noise as measured on the test pulse line (green), and the excess contribution (purple). This breakdown allows observing that the electronic noise is the dominant contribution only for the 5.9 keV line. At medium energies (13.9 keV, and 26.4 keV) the Fano contribution increases significantly, becoming the dominant contribution for the 59.5 keV line. However, on the 122 keV line, the excess contribution becomes by far the most important contribution, showing that the obtained raw resolution is still significantly limited by the transport characteristics of the detector’s material and by the geometrical dimensions of the pixel [24].

Even in presence of the discussed excess contribution, the presented raw spectroscopic results open the path for CZT detectors employable in highly energy-resolved spectroscopic applications without the need for digitally-intensive spectral unfolding techniques [46], which can effectively limit the ultimate count-rate capability in large scanning and imaging applications. Deep sub-microsecond pulse shaping, in ballistic deficit conditions, was also tested validating the capability of the system to be operated in fast, real-time acquisition applications of next-generation ERPC systems, without compromising the spectroscopic and linearity performance of the detection system. A line width of 850 eV FWHM at 59.5 keV was measured at 32 ns peaking time, recording for the first time a spectroscopy-grade performance with a shaped signal’s occupation time compatible with input count-rates in the Mcps region.

As a last remark, the stability over time of the detection system was tested, without resetting the high-voltage detector bias, confirming the presence of a small peak-shift of the 59.5 keV line (<0.45%) and FWHM degradation <35% due to bias-induced polarization effects. Further testing is mandatory to precisely asses flux-induced polarization phenomena that might worsen spectroscopic performance in the Mcps regime.

## Figures and Tables

**Figure 1 sensors-23-02167-f001:**
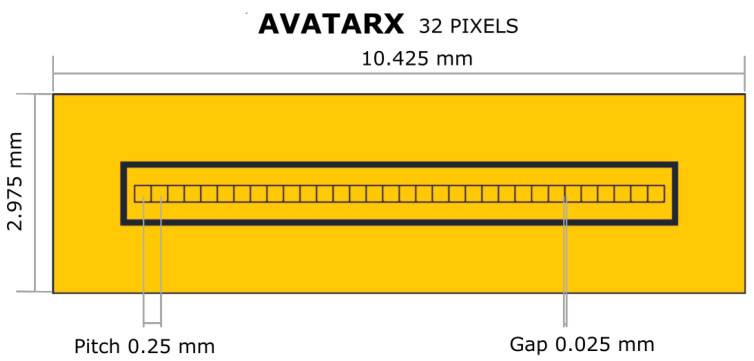
Geometry of the CZT linear array. An inner guard ring is present that can be connected to a dedicated read-out channel to allow charge sharing corrections.

**Figure 2 sensors-23-02167-f002:**
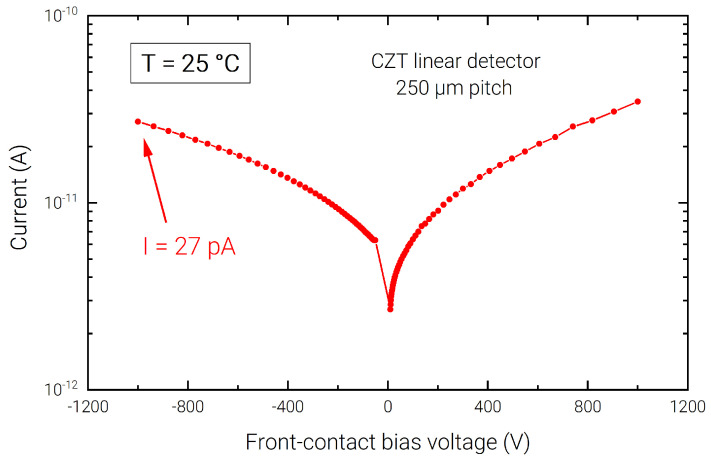
I–V characteristic of a pixel of the CZT linear array at +25 °C.

**Figure 3 sensors-23-02167-f003:**
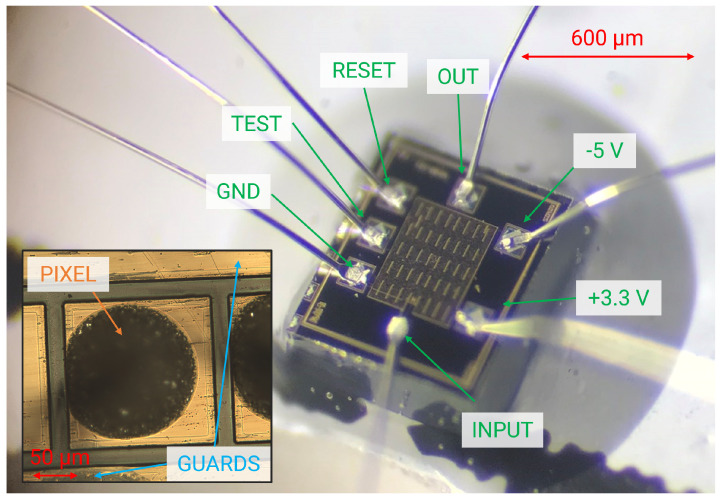
Micrograph of a SIRIO charge sensitive preamplifier (650 × 600 µm^2^) with the bonding-pad pinout. In the inset, a detail of a CZT pixel wire bonded to the preamplifier using a small drop of conductive glue (dark brown circle).

**Figure 4 sensors-23-02167-f004:**
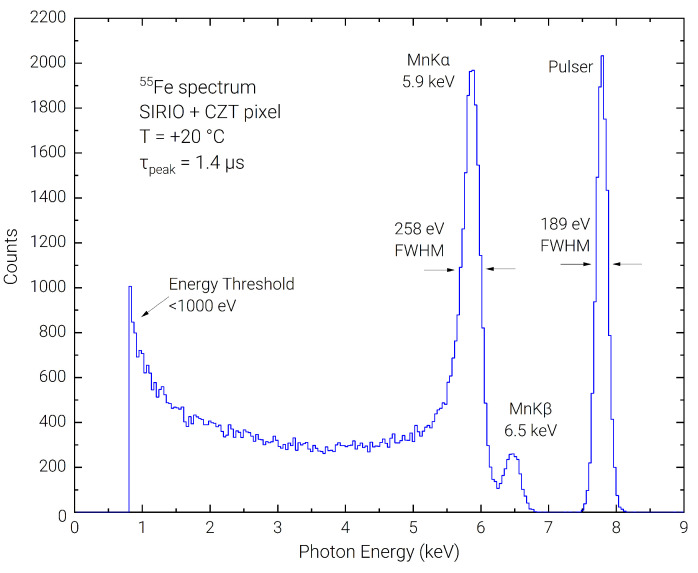
Room temperature energy spectrum acquired with ${}^{55}$Fe calibration source.

**Figure 5 sensors-23-02167-f005:**
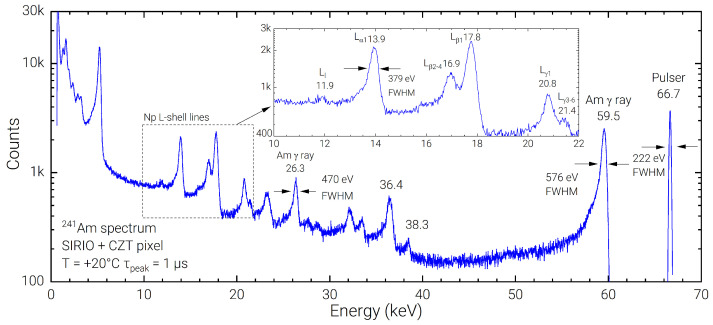
Best room temperature spectrum of a ${}^{241}$Am source acquired using Amptek PX5 DPP at the optimum peaking time of 1 µs.

**Figure 6 sensors-23-02167-f006:**
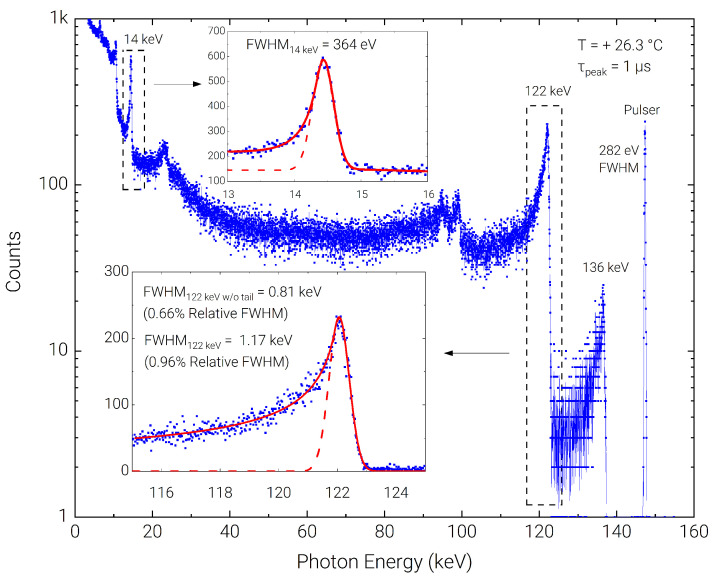
Room temperature spectrum acquired with the ${}^{57}$Co radioactive source. The presence of long left-hand-side tails, as highlighted in the inset of the ${}^{57}$Co is minimized when low-energy photons are absorbed.

**Figure 7 sensors-23-02167-f007:**
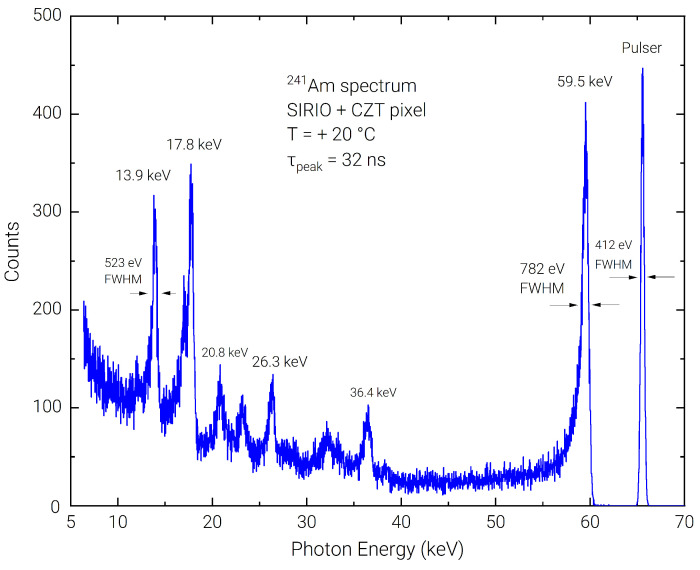
Detail of the room temperature energy spectrum acquired with the ${}^{241}$Am radioactive with 32 ns peaking time.

**Figure 8 sensors-23-02167-f008:**
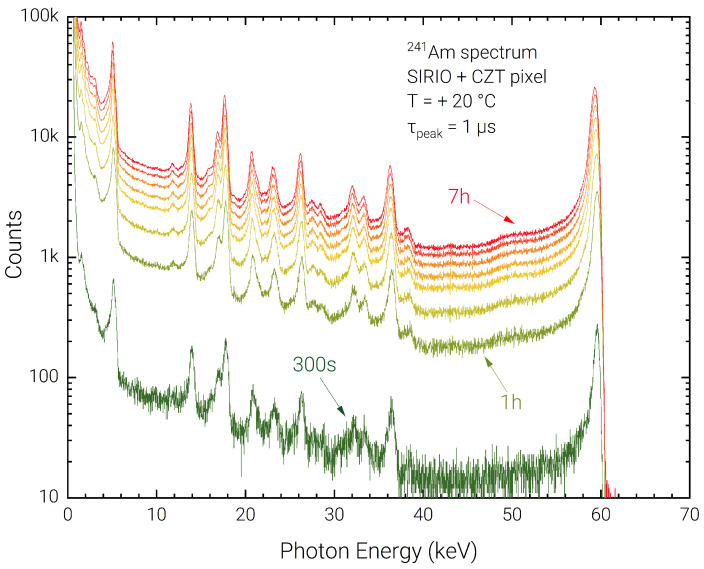
Room temperature energy spectra acquired in a 7-h measurement session, with 1-h acquisition steps.

**Figure 9 sensors-23-02167-f009:**
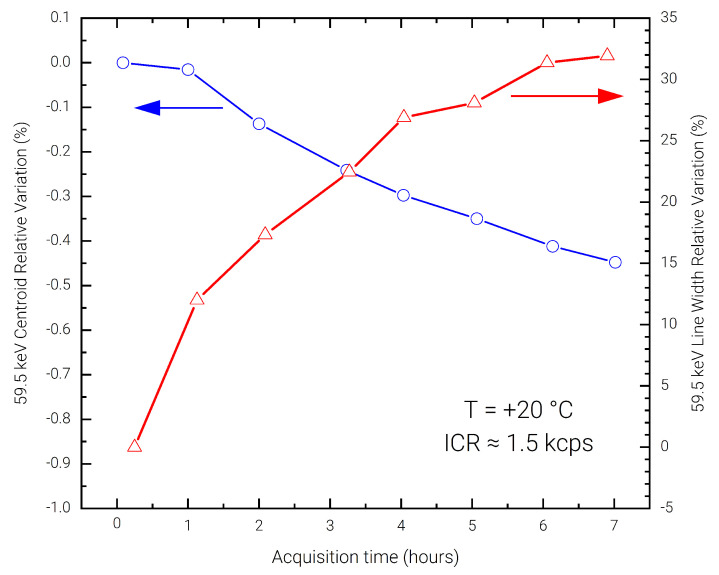
Measured relative centroid shift and FWHM degradation on the 59.5 keV photo-peak over 7-h acquisition time.

**Figure 10 sensors-23-02167-f010:**
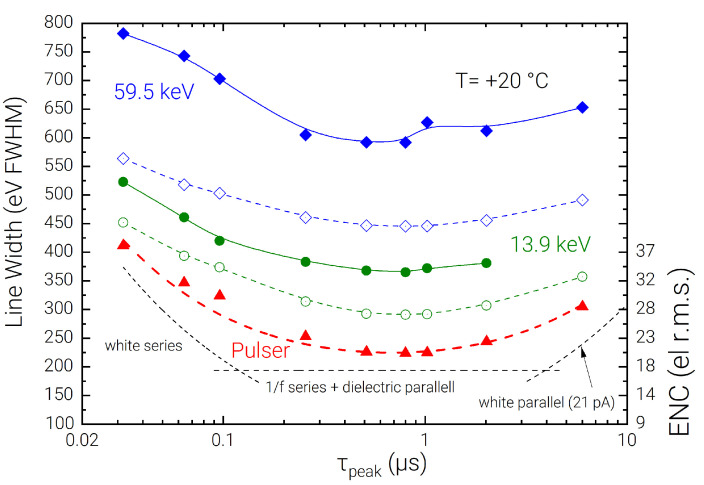
Line widths at room temperature of the SIRIO-CZT system with a ${}^{241}$Am calibration source on the 59.5 keV and 13.9 keV lines with fast peaking-time sweep performed using Dante DPP. Solid blue and green lines are measured FWHM values, while the respective dashed lines are the expected FWHM values summing the pulser FWHM and Fano noise contribution. The estimated series and parallel noise contributions on the pulser FWHM are shown separately with dashed black lines and quadratically summed on the red dashed line. The shot noise associated with the detector current (21 pA) is shown on the bottom-right, estimated evaluating the slope (V/µs) of the ramp at the output of the CSA in absence of incoming photons [45].

**Figure 11 sensors-23-02167-f011:**
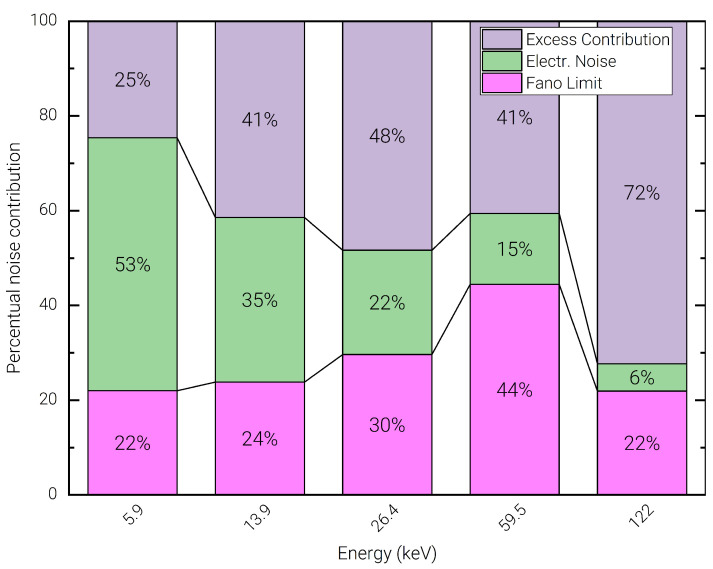
Percent contribution of Fano noise (pink), electronic noise (green), and measured excess contribution (purple) on the line FWHM for different photon energies.

**Table 1 sensors-23-02167-t001:** Summary of measured FWHM in eV (energy resolution in %) and excess contributions, considering a Fano factor of 0.1 [33] and an electron-hole pair generation energy of 4.46 eV [41]. Measurements performed at 1 µs peaking time and +20 °C, except where indicated.

Spectral line	Fano limit FWHM	Pulser + Fano	Measured FWHM	Excess Contribution
5.9 keV (${}^{55}$Fe) ^1^	121 eV (2.1%)	224 eV (3.8%)	258 eV (4.4%)	128 eV (2.2%)
13.9 keV (${}^{241}$Am)	185 eV (1.3%)	290 eV (2.1%)	379 eV (2.7%)	244 eV (1.7%)
14.4 keV (${}^{57}$Co) ^2^	189 eV (1.3%)	339 eV (2.4%)	364 eV (2.5%)	132 eV (0.9%)
26.4 keV (${}^{241}$Am)	256 eV (1.0%)	338 eV (1.3%)	470 eV (1.8%)	327 eV (1.2%)
59.5 keV (${}^{241}$Am)	384 eV (0.6%)	444 eV (0.7%)	576 eV (1.0%)	367 eV (0.6%)
122 keV (${}^{57}$Co) ^2^	549 eV (0.4%)	617 eV (0.5%)	1.17 keV (1.0%)	997 eV (0.8%)

^1^ Measurement performed at 1.4 µss peaking time. ^2^ Measurement performed ad +26.3 °C.

## Data Availability

Not applicable.

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
