# Peer review of "Advances in High-Energy-Resolution CdZnTe Linear Array Pixel Detectors with Fast and Low Noise Readout Electronics"

_sensors, 2023, doi:10.3390/s23042167_

Round 1

Reviewer 1 Report

Minor revisions / suggestions for improvements:

L. 5: this sentence conveys the message that other room-temp. semiconductor detectors have, w.r.t. CZT, higher energy resolution. I thought all the classical high-Z materials have comparable energy resolution capabilities. I this false? If not, can the sentence be corrected?

L. 6 and L. 34: it would be beneficial if the "low energies" were quantified. The reader may not be aware of what it means in this context. Moreover, for low-energy applications, why a high-Z sensor is needed? Can this be explained in the text?

L. 40-43: why was CdTe better than CZT? Is it only due to the readout electronics or there is some physical motivation? Can this be better highlighted in the text?
Further: what happens if one uses the SIRIO electronics on a CdTe sensor? On a higher level, I mean: is the use of CZT mandatory to achieve such resolutions or not?

L. 44: I would mention the pixel pitch here instead of the size of the metal. By the way, is there any particular reason behind the choice of such pixel size?

Fig.3: It would be beneficial if in the photograph you could place a scale reference and some annotations on what is what.

L. 60: Is there any particular reason behind the choice of -700 V as bias voltage? Higher biases can give less charge sharing and tailing and therefore the spectra could be even better, without giving appreciable increase of the noise, according to Fig. 11.

L. 66: Is the input capacity of the SIRIO front-end anyhow matched with the one of the sensor pixel? Is the sensor AC or DC coupled?

L. 98: Can you mention that the radiation is impinging on the uniform (non-pixelated) side of the sensor in Sec. 2?

L. 110-112: Charge collected by neighboring pixels would induce bipolar signals with net area 0 on the pixel under test. With a filter peaking time of 1 us I would expect a complete filtering of those signals. Do you agree? If so, I would ask to remove this explanation.

Fig. 6: from the figure it is clear that FWHM w/o tail is computed with some Gaussian fitting of the peak. It is less clear how the FWHM w/ tail is extracted. Since the whole work focus on the energy resolution, can you explain the fitting procedure in Sec. 2?

Fig. 7: I find very hard to track the changes in the spectra with time. Would a collapsed 2D view help the visualization?

L. 137: Fig. 9 is mentioned before Fig. 7 and 8.

L. 163: Fig. 11 is mentioned before Fig. 10.

Fig. 11: In the caption, the red curve is not mentioned. Further, can it be explained how the contribution of the detector leakage current was estimated?

Reviewer 2 Report

Dear Authors,

I believe that the results on the spectral lines are very good and deserve publication. However, some critical information is currently missing from the manuscript and needs to be included in order to put the work in the appropriate context and allow less experienced reader to appreciate the results and realize the potential limitations.

On the sensor, the authors should provide, at minimum, information on the pixel capacitance. In Figure 1 it would be beneficial to point out to the two guard rings. The authors should include, in the abstract, the information on the pixel size, thickness and applied voltage before reporting the results on the spectral lines. Additionally, the authors should specify in the manuscript if the sensor is illuminated from the anode or cathode side, clarifying why.

On the SIRO front-end it would be beneficial to include a brief description of the electronics chain and, more important, specify the die size and power dissipation. Figure 2 is very poor and doesn’t add information. Rather, the authors should use a microphotograph of SIRIO wire-bonded to the pixel.

In section 2 the authors report, about SIRIO, an “optimization for minimum noise”. This statement is misleading or needs to be clarified. It is well known that the front-end optimization is done with respect to the targeted detector and not in terms of absolute noise.

In section 2 the authors indicate the use of a trapezoidal shaper with a 50ns flat top. However, information on the rise/fall time should also be given, along with the definition of “peaking time” for a trapezoidal shaper. For example, depending on the definition, a peaking time of 35ns seems in contradiction with the use of a 50ns flat top. Also, the authors should clarify why a 50ns flat top has been adopted, and if the choice is related to the charge induction time.

In the Discussion section the authors should discuss the contributions to the electronics noise (i.e. series and parallel). Also, the term “excess noise” doesn’t seem appropriate, given the fact that factors beyond noise may contribute to the FWHM broadening. More suitable terms would be “excess contribution” or “excess broadening”.

Round 2

Reviewer 2 Report

Thank you for addressing my concerns.